# Genome Sequence Analysis of First Ross River Virus Isolate from Papua New Guinea Indicates Long-Term, Local Evolution

**DOI:** 10.3390/v13030482

**Published:** 2021-03-15

**Authors:** Alice Michie, John S. Mackenzie, David W. Smith, Allison Imrie

**Affiliations:** 1School of Biomedical Sciences, University of Western Australia, Nedlands, WA 6009, Australia; alice.michie@uwa.edu.au; 2PathWest Laboratory Medicine Western Australia, Nedlands, WA 6009, Australia; j.mackenzie@curtin.edu.au (J.S.M.); david.smith@health.wa.gov.au (D.W.S.); 3School of Chemistry and Molecular Biosciences, University of Queensland, St. Lucia, QLD 4072, Australia; 4Faculty of Health Sciences, Curtin University, Bentley, WA 6102, Australia

**Keywords:** arbovirus, mosquito-borne disease, alphavirus, Papua New Guinea, Australia

## Abstract

Ross River virus (RRV) is the most medically significant mosquito-borne virus of Australia, in terms of human morbidity. RRV cases, characterised by febrile illness and potentially persistent arthralgia, have been reported from all Australian states and territories. RRV was the cause of a large-scale epidemic of multiple Pacific Island countries and territories (PICTs) from 1979 to 1980, involving at least 50,000 cases. Historical evidence of RRV seropositivity beyond Australia, in populations of Papua New Guinea (PNG), Indonesia and the Solomon Islands, has been documented. We describe the genomic characterisation and timescale analysis of the first isolate of RRV to be sampled from PNG to date. Our analysis indicates that RRV has evolved locally within PNG, independent of Australian lineages, over an approximate 40 year period. The mean time to most recent common ancestor (tMRCA) of the unique PNG clade coincides with the initiation of the PICTs epidemic in mid-1979. This may indicate that an ancestral variant of the PNG clade was seeded into the region during the epidemic, a period of high RRV transmission. Further epidemiological and molecular-based surveillance is required in PNG to better understand the molecular epidemiology of RRV in the general Australasian region.

## 1. Introduction

In Australia, the most common mosquito-borne viral infection is caused by Ross River virus (RRV), an alphavirus of the family *Togaviridae* [1]. Disease associated with RRV infection is characterised by potentially persistent and debilitating fatigue, myalgia, and arthralgia and/or arthritis. RRV cases have been reported from all Australian states and territories since it was first isolated from *Aedes vigilax* mosquitoes sampled in far-north Queensland in 1959 [2].

RRV is currently considered to be both a mosquito vector species and amplifying host generalist, with macropods (such as wallabies and kangaroos) considered the major reservoir host for RRV maintenance and transmission [3]. Humans have been implicated as short-term amplifying hosts during disease epidemics including the large-scale, virgin-soil epidemic that occurred across multiple Pacific Island countries and territories (PICTs) from 1979 to 1980 [4,5,6]. Cases were reported from Fiji, American Samoa, the Cook Islands, New Caledonia and Wallis and Futuna, with disease suspected in northern Tonga [7]. Samoa did not report cases during this time, but during an April 1980 investigation of a dengue outbreak, high titres of anti-RRV haemagglutination inhibition (HI) antibody were detected. RRV activity likely occurred in Samoa during the 1979–1980 outbreak but went undetected [7]. Viraemic travelers are the suspected mode of rapid spread between these adjacent island nations [8].

RRV transmission seemingly ceased in the region in mid-1980, presumably due to the lack of efficient reservoir hosts in the area [7,8]. A viraemic traveler from eastern Australia has been postulated as the source of the outbreak, which began in Fiji in April 1979, based on sequence similarity of epidemic viruses to the ‘Eastern-Australian’ lineage of RRV [9]. The geographical basis of RRV lineage nomenclature was not supported in our recent, more comprehensive genome-scale RRV phylogenetic analysis, where we showed that virus isolates that clustered as distinct lineages were geographically dispersed and sampled Australia wide [10]. The exact origins of the PICTs epidemic is now less certain, given viruses similar to those sampled during the epidemic, were sampled across Australia between 1982 and 1988 [10]. Four distinct genotypes of RRV (G1–4) have been characterised, with G4 in contemporary circulation across Australia [10].

RRV seropositivity has been described in the adult and general population of many regions of Papua New Guinea (PNG), sampled from 1960 to 1969, as well as regions of Indonesia and the Solomon Islands [11]. A later study found a 59% RRV seroprevalence in the Southern Highlands of PNG in 1991, and endemicity was suggested by increasing antibody prevalence with age [12]. RRV as a cause of arthritis in PNG has been confirmed [13,14]. RRV has been isolated from several mosquito species within Australia that are also present in PNG including *Ae. vigilax*, *Ae. notoscriptus* and *Mansonia uniformis* [15,16]. The exact prevalence, distribution, and genetic diversity of RRV in PNG is currently unknown, due to the lack of comprehensive surveillance in the region [17]. RRV cases were not reported from PNG during the PICTs epidemic.

The first and only isolation of RRV from PNG to date was made in 1997 from a pool of *Anopheles farauti* mosquitoes collected at Wandoo village in the Bensbach region of the Western Province [16]. In this study, the genome sequence of this PNG isolate was derived and analysed with available RRV genome sequences to better understand the molecular epidemiology of RRV in the greater Australasian region.

## 2. Materials and Methods

### 2.1. Virus Isolation, RNA Extraction and Next Generation Sequencing

The virus isolate, PNG3075, was isolated from a pool of *An. farauti* mosquitoes trapped in 1997 from the Bensbach locality of the Western Province of PNG [16]. The isolate was identified as RRV using a tissue culture-based fixed cell enzyme immunoassay involving specific, identifying monoclonal antibodies on fixed C6/36 monolayers (*Ae. albopictus* larvae, ATCC^®^ CRL-1660™) [16,18]. The isolate was not investigated on a molecular level at the time of its initial characterisation.

PNG3075 was amplified in the present study through a single passage on Vero cells (African Green Monkey kidney epithelium, ATCC^®^ CCL-81™) maintained in Dulbecco’s Modified Eagle Media (Gibco DMEM, Thermo Fisher Scientific, Waltham, Massachusetts, USA) supplemented (by volume) with 5% heat-inactivated fetal bovine serum (FBS), 1% L-glutamine (Life Technologies, Carlsbad, California, USA) and 1% penicillin/streptomycin (Life Technologies). The Roche High Pure RNA extraction kit (Roche, Basal, Switzerland) was utilised as per the manufacturers protocol from viral supernatant, concentrated through ultra-centrifugation in Millipore Amicon Ultra-15 centrifugal units (Merck Millipore, Darmstadt, Germany).

The TruSeq stranded mRNA kit (Illumina, San Diego, California, USA) was used for library preparation. Superscript III reverse transcriptase (Invitrogen, Waltham, Massachusetts, USA) was used for cDNA synthesis. Libraries were validated using the Agilent 1000 DNA kit (Agilent Technologies, Santa Clara, California, USA) and subsequently normalized and pooled. Sequence data were generated on the MiSeq platform with the MiSeq Micro v2 kit (Illumina). Paired-end reads of 150 bp were demultiplexed and assessed for quality within the FastQC v0.11 program [19], and subject to de novo genome sequence generation within CLC Genomics Workbench v7.5 (QIAGEN, Hilden, Germany) to derive a full-coding contiguous sequence, 11,837 nucleotides in length. A total of 567,724 reads were generated for the PNG3075 sample, of which 86.89% (493,294 reads) were used to generate the complete coding PNG3075 sequence.

### 2.2. Phylogenetic Analysis

The derived genome was aligned with 104 publicly available RRV genome sequences using MAFFT 7.338 within Geneious 11.1.4 [20]. Most available sequence data were derived in our recent RRV phylogenetic investigation, with isolates primarily sampled in Western Australia [10].

For phylogenetic analysis, sequences were manually trimmed to remove 5′ and 3′ untranslated regions (UTRs) and then realigned. A maximum likelihood phylogeny was reconstructed using RAxML 8.2.11 with 1000 bootstrap replicates within Geneious 11.1.4, under a GTR+G+I nucleotide substitution model—the best-suited model according to JModelTest 2 analysis [21,22]. All phylogenies were visualized and edited within FigTree v1.4.4 [23].

### 2.3. Temporal Analysis

The timescale of divergence of major RRV clades was estimated using Bayesian Markov chain Monte Carlo (MCMC) analysis within the BEAST package (v1.8) [24]. The dataset demonstrated considerable temporal signal (R^2^, 0.921; Correlation Coefficient, 0.96) in root-to-tip regression analysis using TempEst v1.5 [25]. An uncorrelated lognormal (UCLN) clock was assumed under a GTR+G+I nucleotide substitution model, and a constant coalescent population with default priors. Three independent chains were run, assessed for convergence with log and tree files subsequently combined using LogCombiner. A maximum clade credibility tree (MCC) was reconstructed with the 105-taxon dataset, with 10% burn-in, and visualized using FigTree v1.4.

### 2.4. Accession Number

The sequence derived in this study was deposited to the NCBI database and assigned the accession number MW238766.

## 3. Results

Genome sequence analysis of isolate PNG3075 confirmed it as RRV, as initially identified through monoclonal antibody analysis when first isolated.

A maximum likelihood (ML) phylogeny of a 105-taxon alignment of RRV complete coding sequences was re-constructed (Appendix A). PNG3075 constituted a unique clade, distinct from all RRV isolates sampled in Australia over a 59 year period (1959–2018, Figure 1). This is consistent, when a phylogeny is constructed from an alignment of a larger dataset of geographically defined partial genome sequences, sampled over the same timeframe (Appendix A). The unique clade is distinct from the genotype 4 (G4) viruses that were in wide Australian circulation at the time of the 1997 sampling of PNG3075. The long-terminal branch of the clade is suggestive of long-term evolution, independent of Australian lineages. Rather than a recent, or continuous exchange of virus between Australia and PNG, this clade was likely introduced into PNG in an historical event with subsequent local evolution. PNG3075 was most similar to isolates K1008 (Kununurra, 1986–genotype 3, G3) and DC7053 (Mandurah, 2005–G4), with 98.9% and 98.6% pairwise sequence identity, respectively.

A 36-nucleotide insertion within the nsP3 gene, that is characteristic of all RRV G3 and G4 isolates, was also present in the PNG3075 genome [10]. The earliest sampled virus containing this insertion, is F9073—the first isolate sampled during the PICTs epidemic, from Fiji in April 1979.

Several RRV isolates have been observed to contain nucleotide insertions or deletions (indels) within the nsP3, of various sizes, mostly affecting one of the four conserved proline motifs of the nsP3 hypervariable domain [10]. No such indels were observed within PNG3075.

To determine the timescale of divergence of the PNG clade from Australian lineages of RRV, the time to most recent common ancestor (tMRCA) was estimated using Bayesian methods, and a maximum clade credibility (MCC) phylogeny re-constructed (Figure 1). An uncorrelated lognormal, relaxed molecular clock was assumed based on results of a general sampling stepping-stone analysis, when compared with a strict clock assumption. The GTR + G + I nucleotide substitution model was employed, with a constant coalescent tree prior.

The median root age of the entire 105-taxon dataset was estimated as July 1924 (mean: 1924.5, 95% highest probability density (HPD) 1904.7–1941.1) under a relaxed clock. The estimated mean nucleotide substitution rate was 3.06 × 10^−4^ (95% HPD 2.62–3.54 × 10^−4^), consistent with previous estimates of alphavirus evolutionary rates [10,26,27]. Each major genotype of RRV, as well as the newly described PNG clade, emerged in approximately decade-long intervals, consistent with our previous genome-scale investigation of RRV [10].

The mean tMRCA of the PNG clade was estimated as July 1979 (mean 1979.5, 95% HPD 1972.6–1984.9), corresponding to the initiation of the PICTs epidemic in Fiji (April 1979) and subsequent spread to neighbouring islands in the region (Figure 2). The mean tMRCA estimate of isolates sampled close to the conclusion of the PICTs epidemic, from the Cook Islands, is comparable to that of the PNG clade estimates (mean 1979.9, 95% HPD 1979.4–1980.0), though with a narrower 95% HPD interval.

Our temporal estimates suggest that an ancestral variant of the PNG clade may have been seeded into PNG during the PICTs epidemic, a period of high RRV transmission in the general Pacific region. It is not known whether virus was seeded to PNG directly from the PICTs at the height of the outbreak, or from Australia immediately before, after, or concurrently with the epidemic. RRV activity in PNG prior to the PICTs epidemic is evident from several historical seroprevalence studies, but molecular information from this period is lacking. Seroprevalence data from 1960 to 1969, encompassing multiple regions in PNG, predate the temporal range of this clade, suggesting that one or more distinct RRV lineage(s) has circulated in PNG and infected inhabitants, in decades prior to the introduction of the PNG clade. Three cases of RRV-induced polyarthritis were documented in the capital, Port Moresby between 1980 and 1981, potentially caused by strains related to the RRV clade described here [13].

## 4. Discussion

Our analysis of the sole RRV isolate to be derived from PNG thus far has shed light on the molecular epidemiology of this medically significant arbovirus in the greater Australasian region. Phylogenetic analysis of the derived genome sequence and 104 available RRV full genome sequences revealed that the PNG isolate constitutes a unique genetic group, distinct from all other Australian RRV lineages. The clade is a result of long-term, local evolution of RRV within PNG, independent of Australian variants.

The establishment and continuation of RRV transmission in PNG may have been facilitated by one or more of the many endemic marsupial species present in the country [28]. A diversity of mosquito species is present in PNG, many of which have been the source of isolations in field-caught specimens in Australia [16,29]. Differences in vector competence for RRV between species present in Australia and PNG, is not known. *An. farauti*, the mosquito species from which the PNG isolate was derived, is present in brackish habitats in northern Australia, but has not been associated with RRV transmission in Australia, as of yet.

With this new sequence from PNG, the current RRV phylogeny has been expanded. Consistent with our previous analysis of RRV, we estimate that major RRV genetic groups, including the PNG clade, arise approximately every decade. The mean nucleotide substitution rate and tMRCA of the entire RRV dataset, was also consistent with the prior RRV analysis [10].

The mean estimated emergence time of the PNG clade was July 1979, corresponding to peak disease reporting during the PICTs RRV outbreak. The PICTs epidemic was previously thought to have been initiated by an eastern Australian traveler based on the sampling of ‘Eastern’ Australian RRV lineage isolates during the outbreak. The exact origins of the PICTs epidemic viruses, which cluster within G3, are now unclear, as our recent analysis [10] showed G3 viruses were sampled throughout Australia within a 6 year period (1982–1988).

RRV disease was not documented in PNG until after the PICTs epidemic, in a 1980 survey of acute arthritis, where 13.6% (of 44 patients) of reported infectious arthritis cases were serologically attributed to RRV infection [14]. Arboviral surveillance, including RRV surveillance, is and has been lacking in PNG with the exact disease epidemiology of RRV within the country, uncharacterised. Serological evidence of RRV was identified in a large proportion of the adult and general population in multiple regions across PNG, in a collection sampled from 1961 to 1969 [11]. This sampling period pre-dates the temporal range estimated obtained in this analysis, by several decades and suggests a separate clade or lineage was in circulation and infecting residents prior to the emergence of the PNG clade. Indeed, if a novel lineage of RRV had been seeded into PNG during the PICTs epidemic, it is feasible that cases went unnoticed, given the potentially high seroprevalence of anti-RRV antibodies in the PNG population. Molecular information of RRV, including the diversity of lineages in past and present circulation, is lacking.

Barmah Forest virus is an alphavirus that was thought to be exclusively endemic to Australia, until its recent isolation from a febrile patient in PNG with no history of international travel [30]. Temporal analysis dates the emergence of the unique BFV PNG clade to 50–120 years ago, likely facilitated by movement of large numbers of Australian troops into the Asia Pacific region during World War 2 [31]. Consistent with our findings for RRV, it appears that exchange of BFV between Australia and PNG is sporadic, despite the theoretical potential for more regular spread via mobile vertebrate hosts such as migratory birds [30,31].

Further molecular surveillance of RRV circulating in PNG would be informative to further clarify the molecular epidemiology of RRV in the wider Australasian region, and to characterise the diversity of viruses circulating in the country, and associations to Australian lineages. This analysis has highlighted the long-term circulation of RRV beyond Australia. The potential of RRV to become established in regions beyond the known range of activity, in areas where supportive mosquito vectors and vertebrate hosts are present, should be considered. Indeed, there is need for a collaborative surveillance system between Australia and its regional partners to facilitate on-going, long-term surveillance of RRV and other medically important arboviruses.

## Figures and Tables

**Figure 1 viruses-13-00482-f001:**
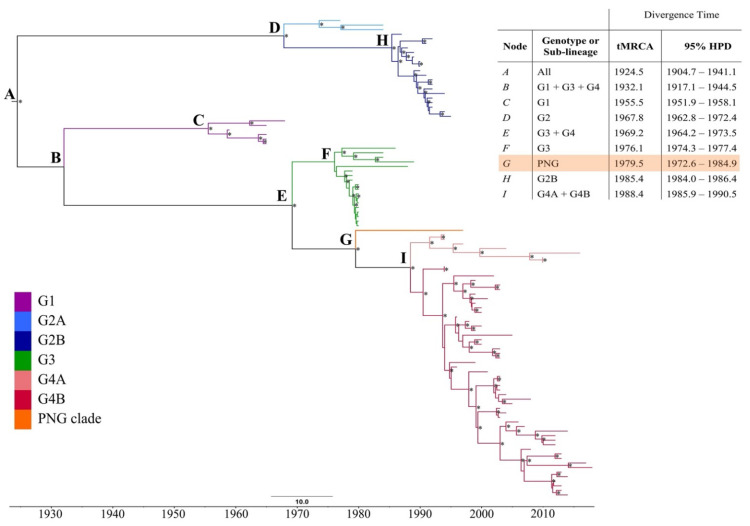
Maximum clade credibility (MCC) phylogeny of the 105-taxon Ross River virus (RRV) dataset. Clades are coloured for the genotype or sub-lineage as characterised by the maximum likelihood phylogeny of Appendix A (see key in lower left). Posterior probability values of > 0.70 are presented above nodes, as indicated by asterisks (*). The table in the upper-right corner presents the mean time to most recent common ancestor (tMRCA) with error reported as the 95% highest probability density (95% HPD) for each major genetic group of RRV. The PNG clade has been highlighted within the table.

**Figure 2 viruses-13-00482-f002:**
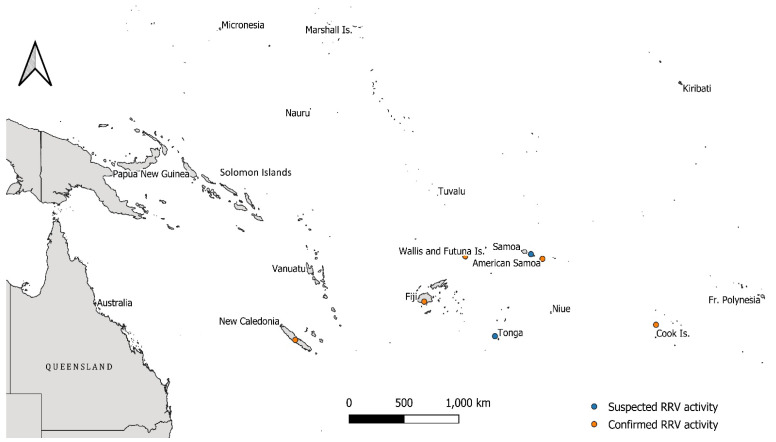
Map of the Pacific region and the location of Pacific Island Countries and Territories (PICTs) involved, or neighbouring, the explosive Ross River virus (RRV) epidemic that occurred between 1979 and 1980. Orange dots highlight PICTs where confirmed RRV activity with virus isolations was documented, while blue dots highlight locations where RRV activity was suspected during the outbreak. The proximity of Papua New Guinea to these areas of activity is of note. The scale bar is shown in the lower-centre of the map.

## Data Availability

The sequence derived in this study was deposited to the NCBI database and assigned the accession number MW238766.

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
