# Peer review of "Genome Sequence Analysis of First Ross River Virus Isolate from Papua New Guinea Indicates Long-Term, Local Evolution"

_viruses, 2021, doi:10.3390/v13030482_

Round 1

Reviewer 1 Report

This study documents the genomic characterisation and analysis of the first isolate of RRV to be sampled from PNG. The introduction provides a useful summary of the RRV PICTs epidemic that occurred during 1979 - 1980. The analysis shows the PNG RRV isolate forms its own genetic group, indicative of long-term, local evolution in PNG. This study is important in highlighting the lack of genetic information for RRV and other arboviruses in PNG and helps justify the need for arbovirus surveillance in this region. Minor edits and suggestions are outlined below. 

  • Consistent abbreviation of the mosquito genus names is needed (e.g. Aedes in Line 62 can be abbreviated to Ae. as this species has already been mentioned; the same is true for Anopheles in Line 73).
  • Lines 92-93: please state the size and type of read used.
  • Lines 95-96: what was the length of the contiguous sequence? Perhaps provide the length of the PNG3075 sequence in the results. 
  • Lines 101-102: remove the words 'and aligned'
  • Lines 122-124: has the PNG3075 sequence deposited to NCBI been made public yet? The accession number MW238766 doesn't lead to anything so I can't check the entry.
  • Figure 2: perhaps decrease the size of the 'QUEENSLAND' font as it is much larger than the font used for the country names, which are of greater importance.
  • Lines 191-192: inconsistent hyphenation of the word 'seroprevalence'. 
  • Lines 207-208: please provide a reference for this statement. 

Author Response

We would like to thank the reviewer for their time and very constructive feedback. The issues raised have been addressed and summarised below:

  • Consistent abbreviations have now been used with Aedes and Anopheles abbreviated to Ae. and An. where appropriate.
  • Paired-end 150bp reads were produced – this information has now been added to the methods section in line 94.
  • An 11,837 nucleotide contig was generated – this information has now been added to the methods section in line 97.
  • The duplicate ‘and aligned’ phrase has been removed.
  • The sequence has been deposited to NCBI. We will make this sequence public once this manuscript is published. 
  • The font size of ‘Queensland’ has been reduced in Figure 2.
  • Inconsistencies with the hyphenation of ‘seroprevalence’ and ‘seropositivity’ have now been amended throughout the manuscript, with the removal of hyphens.
  • Two references have been added to the statement made in line 209, in regard to mosquito species present in PNG and Australia (Johansen et al. 2000 and LaPointe e2007.

Reviewer 2 Report

This article written by Michie et al. described the first genome sequence of RRV (Ross River virus) from Papua New Guinea. Authors presented clear and sufficient introduction about distribution of RRV in Australia and Pacific island countries, its outbreaks and information about serological data in human population and molecular screening in mosquitoes. The part Material and Methods is described in detail with the information about every step in the process of getting genome sequence of virus isolate PNG3075. This new genome sequence is later compared with other sequences derived from this region and Australia. This RRV virus from PNG formed unique clade within other RRV virus sequences. They also focused on timescale divergence of new PNG sequence with clear local evolution of RRV in PNG outside of Australia.

The manuscript is well written and I have no comments.

Question:

Dear authors, you focused on sequencing of virus isolate of RVV obtained in 1997 from PNG. You got complete genome sequence that you compared with other sequences from Australia. I have one question regarding situation about RRV in Australia, Pacific island countries and PNG now. Have you observed some epidemic in recent years? Have you noticed some RRV infection in humans in Australia or in Pacific island or in PNG in recent years? What is the main driver of increasing human cases caused by RVV, some weather condition, overpopulation of mosquitoes, travelling of viraemic patients, etc.? What is the situation of RVV in PNG now? Is it well established virus?

Author Response

We would like to thank this reviewer for their comments and feedback. To answer the reviewer’s question, the epidemiology of RRV in PNG and islands of the Pacific is currently unclear due to the lack of surveillance. There has not been a large-scale epidemic of RRV reported in the Pacific since the 1979-1980 outbreak reported in this manuscript. Human RRV cases are reported throughout Australia every year, with an average of 5000 cases reported annually. There have been over 1000 RRV cases reported so far in 2021, nation-wide. The factors influencing RRV activity are complex, but certainly include climatic factors and other factors that influence vector abundance and availability of susceptible amplifying hosts. There have been recent publications citing evidence of RRV activity in Fiji, American Samoa and French Polynesia based on serosurveys, though there have been no isolations of virus made since the 1979-1980 outbreak.